# The Host Non-Coding RNA Response to Alphavirus Infection

**DOI:** 10.3390/v15020562

**Published:** 2023-02-18

**Authors:** Mahgol Behnia, Steven B. Bradfute

**Affiliations:** Department of Internal Medicine, University of New Mexico Health Sciences Center, Albuquerque, NM 87131, USA

**Keywords:** RNA, noncoding, miRNA, lncRNA, snoRNA, alphavirus, encephalitis, chikungunya

## Abstract

Alphaviruses are important human and animal pathogens that can cause a range of debilitating symptoms and are found worldwide. These include arthralgic diseases caused by Old-World viruses and encephalitis induced by infection with New-World alphaviruses. Non-coding RNAs do not encode for proteins, but can modulate cellular response pathways in a myriad of ways. There are several classes of non-coding RNAs, some more well-studied than others. Much research has focused on the mRNA response to infection against alphaviruses, but analysis of non-coding RNA responses has been more limited until recently. This review covers what is known regarding host cell non-coding RNA responses in alphavirus infections and highlights gaps in the knowledge that future research should address.

## 1. Introduction

The genus Alphavirus in the *Togaviridae* family represents a diverse group of small, enveloped RNA viruses comprising 32 species in 8 antigenic complexes [1,2]. The spherical virion of alphaviruses is ~70 nm in diameter and comprises the encapsidated single-stranded RNA genome [3]. The positive-sense single-stranded RNA genome of alphaviruses ranges from 11 and 12 kb in length, with a cap structure at the 5′ end and a poly-A tail at the 3′ end. The genome consists of two open reading frames (ORFs); one is flanked by a 5′ cap and an untranslated region which encodes four nonstructural proteins, and the other which is controlled by the subgenomic promoter and encodes five structural proteins [4,5]. Having mRNA characteristics, the genome of alphaviruses initiates the viral replication process upon introduction to susceptible cells and, as a result, is known as infectious.

Alphaviruses are found on all continents [6]. A recent study suggested that the emergence of alphaviruses from marine animals in the Southern Hemisphere’s oceans led to their subsequent spread to terrestrial hosts [7]. Alphaviruses are arthropod-borne, and mostly mosquito-borne, which makes them a recurring threat to human health as the expansion of mosquito habitats may put more people at risk of exposure to these viruses. In nature, alphaviruses maintain a sylvatic cycle between hematophagous arthropod vectors and small vertebrate hosts. However, the occasional spillover from these enzootic cycles results in human infection by alphaviruses. Based on the disease caused in humans during natural infection, alphaviruses are grouped into New-World and Old-World alphaviruses [8]. The New-World alphaviruses, including Venezuelan equine encephalitis virus (VEEV), Western equine encephalitis virus (WEEV), and Eastern equine encephalitis virus (EEEV), cause encephalomyelitis in humans and are mostly found in the Americas, whereas their Old-World relatives, including Sindbis virus (SINV), Simliki Forest virus (SFV), chikungunya virus (CHIKV), and Ross River virus (RRV), cause polyarthritis and are found in Asia, Africa, Europe, and Oceania [9]. Although most alphaviruses have vertebrate hosts, some such as Eliat virus, Taï Forest alphavirus, and the recently isolated Agua Salud alphavirus are restricted to the insects and their replication in vertebrate hosts has not been reported [10,11,12].

The human mortality rate post-infection with alphaviruses differs between New-World and Old-World alphaviruses. The mortality rate in Old-World alphaviruses, such as CHIKV, is very low (~0.03%), whereas the mortality rate in some New-World alphaviruses, such as EEEV, is 50–75% [5]. Although most alphaviruses cause acute human diseases, many patients suffer from chronic debilitating sequelae following infection with these viruses. This includes long-lasting arthralgia and myalgia following infection with Old-World alphaviruses or life-lasting neurological disorders following New-World alphavirus infection [13,14]. Currently, no approved vaccine or antiviral treatment is available against the alphaviruses for human use, emphasizing the importance of investigating the host–alphavirus interaction to find possible targets for therapy.

Recent advances in sequencing technologies and more in-depth transcriptomic analysis revealed that at least 76% of the human genome is transcribed, from which less than 2% is made up of protein-coding genes [15,16]. The terminal product of the remaining portion of the transcribed genome, previously regarded as “dark matter,” is non-coding RNA (ncRNA). ncRNA is a class of transcripts with no to little translation potential [17,18,19,20,21], which is classified into two major groups, housekeeping and regulatory ncRNAs, based on their respective roles in biology. The regulatory ncRNAs are classified based on their length into short and long ncRNA groups. The short ncRNAs are <200 nucleotides (nt) in length. This group includes microRNA (miRNA), endogenous small interfering RNA (siRNA), and P-element-induced wimpy testis (PIWI) interacting RNA (piRNA), which are involved in post-transcriptional regulations of gene expression. Of short ncRNAs, miRNAs are the most studied, and their roles in many cellular processes, including development, proliferation, and apoptosis, have been well-described [22,23,24,25,26].

Long non-coding RNA (lncRNA) is a heterogeneous group of ncRNAs that are >200 nt in length, and differ in their biogenesis and genomic origin. Recent transcriptomic analysis has reported the presence of 30,000 to 60,000 lncRNAs in the human transcriptome [27,28]. Although called non-coding, improvement in bioinformatics and high-throughput sequencing has revealed the presence of short open reading frames (ORFs) in certain lncRNAs, which can be translated to short peptides [21,29]. Unlike short ncRNAs, the cellular roles of many lncRNAs have not been investigated. However, the more studied lncRNAs are shown to regulate gene expression at multiple levels. By interacting with DNA, RNA, or proteins, lncRNAs regulate the chromatin state, which affects the transcription of the nearby and distant genes. They can also affect gene expression by modulating RNA splicing, stability, and translation [30].

Changes in the expression of non-coding genes have been reported in many human diseases [31,32,33]. In the past decade, an increasing number of studies in the literature have reported on the modulation of host ncRNAs in response to viral infections and illustrated their anti- and proviral roles. Many host ncRNAs exert their antiviral effect by activating antiviral signaling pathways, which affect viral replication and infection. On the other hand, some viruses hijack or sequester host ncRNAs as a mechanism for boosting their replication or evading the immune response. Besides host ncRNAs, viral-encoded ncRNAs are also reported to modulate viral infection. For example, the production of viral-encoded miRNAs (v-miRNAs) and lncRNAs and their roles in viral infection have been reported by multiple groups [34,35]. In this review, we summarize the current knowledge about differential host ncRNA expression and their role in alphavirus infection. 

## 2. Roles of miRNAs during Alphavirus Infection

RNA interference (RNAi) is a post-transcriptional mechanism of gene silencing conducted by short ncRNAs. miRNAs are an essential arm of RNAi that negatively regulate gene expression through binding to complementary sequences in the target mRNA, resulting in translation blockage or RNA decay [36,37]. Although, in most cases, the miRNA’s interaction with the 3′ untranslated region (UTR) of the target mRNA results in lower transcript or protein levels, there are reports of miRNAs interacting with the 5′UTR, promoter, or coding sequence in the mRNA, some of which resulted in translation activation [38]. Through the regulation of gene expression, miRNAs play critical roles in important cellular processes, including development, cell proliferation and death, hematopoiesis, and immune function [39,40]. The RNAi machinery modulates viral infection through different mode of actions: (A) Viral infection alters the host miRNA profile. The differentially expressed miRNAs either interact with host transcripts to modulate the immune response to the infection, or with the viral genome to modulate viral replication [41]. (B) The viral genome is processed by RNAi machinery to produce viral miRNA, which targets host gene expression and alters the cellular response to the infection [35]. (C) RNAi produces small viral RNAs and uses them as a template to target the viral genome [42]. Therefore, RNAi can play anti- or proviral roles in viral infection. This section discusses the roles of the host miRNAs in alphavirus infection.

### 2.1. Antiviral Role of miRNAs in Mammalian Cells

Many viruses, including alphaviruses, have strategies to evade the cellular immune response to establish an effective infection [43,44,45]. Conversely, the host cells activate genes and pathways to defend themselves against viral infection. Therefore, in order to survive, both host cells and viruses alter the cellular microenvironment using different mechanisms, e.g., modulations of non-coding RNA expression, during the infection. 

The host miRNA profile is known to be altered in many viral infections [35]. Multiple studies have investigated the miRNA signature during alphaviruses, specifically during CHIKV infection, and identified the involvement of miRNAs in modulations of host immune pathways during infection. A study of the miRNA signature in CHIKV-infected human and mouse skin fibroblasts demonstrated the upregulation of eleven miRNAs and the downregulation of five miRNAs. Further gene ontology and pathway analysis showed immune-related pathways such as RIG-I, JAK-STAT, MAPK, TGF-β, cytokine-cytokine receptor, and Fc gamma-mediated phagocytosis signaling pathways as the targets of modulated miRNAs post-CHIKV infection in these cells [46]. Furthermore, the modulation of TNF, TLR, MAPK, IL-6, and AKT signaling pathways by differentially expressed miRNAs have been observed in CHIKV-infected primary human synovial fibroblasts [47]. CHIKV infection has been shown to elicit a strong IFN type I response, which results in the elevated expression of proinflammatory mediators in response to the infection. The strong tropism of CHIKV to muscle, joint, and dermal fibroblasts and the replication of the virus in them [48] leads to enhanced type I IFN responses and the secretion of proinflammatory mediators, resulting in the infiltration of immune cells into the tissue and inflammation [49]. The inflammation in the infected tissue aids in viral clearance. but also contributes to persistent CHIKV-related arthralgia and inflammation [50]. Therefore, modulation of the immune-related pathways in fibroblasts may affect the outcome of the CHIKV infection. Notably, the perturbation of TLR and JAK-STAT signaling pathways and the host immune response has been reported in alphavirus infections, including in CHIKV infection [45,51,52,53,54]. Although the immune-related signaling pathways in CHIKV infection were mostly the target of upregulated miRNAs, the elevated expression of miRNAs does not necessarily result in the suppression of the target genes [55]. Therefore, the study of the role of these miRNAs in modulations of the immune response is essential for understanding the host–pathogen interactions. 

Microarray analysis has shown modulations of 152 miRNAs in CHIV-infected HEK293T cells, and predicted TGF-β, Wnt, and adherent junction signaling pathways as targets of these miRNAs [56]. Among the modulated miRNAs, 65–70% were upregulated, and the rest were downregulated. Interestingly, 53% of these upregulated and 45% of these downregulated miRNAs have been implicated in HCV, HBV, HPV, and HIV1 infections, suggesting the possibility of a common antiviral response to viral infections. Further validation of the microarray gene expression using qPCR confirmed the induction of miR-744, miR-638, and miR-503 among the highly expressed miRNAs [56]. miRNA clusters consist of two or more adjacent miRNA genes that are jointly transcribed as polycistronic miRNAs [57]. The additional study of the modulated miRNAs post-CHIKV infection in HEK293T cells has shown that upregulated miRNAs belong to several miRNA clusters, including miR-17-92, let7e/99, miR-191/425, miR-106b/miR25, mi-R23a/24, and miR15b/16, indicating the co-regulation of miRNA clusters in response to CHIKV infection [56]. Of note, miR-23a, miR-106b, and miR-17-92, members of upregulated miRNA clusters in CHIKV-infected HEK293T cells [56], are regulators of the TGF-β pathway [58,59], and their elevated expression was also reported in mammalian fibroblasts during CHIKV infection [46]. Interestingly, the qPCR assessment of mRNA expression confirmed the strong induction of genes involved in TGF-β signaling (SMAD6, JUN, and SKIL) in both HEK293T cells and primary human dermal fibroblasts, showing the high expression of miR-23a, miR-106b, and miR-17-92 during CHIKV infection [56]. In addition, miRNA–mRNA crosstalk analysis showed the TGF-β pathway as the target of downregulated miRNAs in CHIKV-infected HEK293T cells and mammalian skin fibroblasts [46,56]. Altogether, these results suggest that miRNAs probably regulate the upregulation of TGF-β in CHIKV infection. TGF-β signaling induces apoptosis [60,61] and is involved in the ingestion of apoptotic cells by macrophages in the immune defense process against viral infection [62]. Multiple viruses have been shown to suppress TGF-β signaling to limit TGF-β-induced apoptosis and impair viral clearance [63]. As fibroblasts are the main targets of CHIKV, the increased TGF-β post-CHIKV infection may indicate a possible host cell response to infection, leading to increased apoptosis of these infected cells and virus clearance.

The study of the miRNA profile in CHIKV-infected primary human synovial fibroblasts revealed the elevated expression of miR-1264 after CHIKV infection and predicted tripartite motif-containing protein 26 (TRIM26) as its target [47]. TRIM26 suppresses IFNβ and the antiviral response through polyubiquitination and degradation of interferon regulatory factor 3 (IRF3) [64,65]. The higher expression of TRIM26 has been reported in viral infections, which was correlated with a suppressed IFN response and higher viral titers. Furthermore, suppression of TRIM26 resulted in lower viral titers [64,65]. The elevated expression of miRNAs is usually correlated with the suppression of the target genes. Therefore, the higher expression of miR-1264 probably suppresses the TRIM26, which results in a higher IFNβ response and lower viral titer. Concordantly, the rapid induction of type I IFN response in fibroblasts, including the expression and secretion of IFNβ and proinflammatory cytokines, has been reported as a part of the cellular innate immune response to CHIKV infection [66]. Thus, the elevated expression of miR-1264 results in lower viral replication in the main target cells of the virus.

Modulation of miRNAs and their target pathways has also been reported in the New-World alphaviruses. New-World alphaviruses, also called encephalitic alphaviruses, can infect the CNS and cause encephalitis. The infection of the cells in the CNS by encephalitic alphaviruses results in an inflammatory response in an attempt to control the viral infection, but can cause brain damage [67,68,69]. An in vivo study of the miRNA alteration pattern in the brain of VEEV-infected CD1 mice demonstrated the significant alteration of the miRNAs involved in the modulation of biological functions such as antigen presentation, inflammation, and apoptosis [70]. The significant expression of miR-155, miR-27a, rno-miR-381, miR-154, and miR-801 has been reported in brains of VEEV-infected CD1 mice, both 48 and 72 h post-infection (p.i.). Further, miRNA target gene analysis predicted proinflammatory cytokine IL-1β as the target of miR-155 [70]. The higher expression of miR-155 has been reported to be associated with higher proinflammatory IL-1β levels and an elevated inflammatory response to viral infection, resulting in a significantly lower viral load in the brain and a higher survival rate in the mouse model of West Nile virus [71]. Although the elevated expression of the proinflammatory cytokines can decrease the viral load, it does not necessarily alleviate the disease condition in all viral diseases. The elevated expression of proinflammatory cytokines, including IL-1β, has been reported in the brains of VEEV-infected mice, which results in increased inflammation leading to neuronal cell death and brain damage [72,73]. Thus, even though the elevated expression of miR-155 in the brains of VEEV-infected mice is a host antiviral response that decreases viral replication by increasing proinflammatory cytokine IL-1β, it may cause severe brain damage due to increased inflammation. 

### 2.2. Proviral Roles of miRNAs in Mammalian Cells

#### 2.2.1. Proviral Modulation of Host Cellular Pathways by miRNAs

miRNAs are major regulators of gene expression in cells. Therefore, many viruses modulate them to change the host cell microenvironment to be in favor of the virus. The study of the miRNA profile post-CHIKV infection has shown downregulation of miRNAs targeting the endocytosis pathway in HEK293 cells. Further qPCR assessment of mRNA expression confirmed the elevated expression of genes involved in the endocytosis (CXCR4, HSPA8, and ADRB1) pathway in CHIKV-infected HEK293T cells and human primary dermal fibroblasts [56]. Many viruses, including alphaviruses, use endocytosis as the mechanism to enter the cells [74]. Therefore, these results may show the proviral role of miRNAs in CHIKV infection.

The microarray profiling of primary human synovial fibroblasts showed upregulation of a subset of 26 differentially expressed miRNAs (DEMs) during the infection with CHIKV. The miRNA–mRNA interaction network showed the highest degree of mRNA interaction (a miRNA with more than six target mRNAs) for four DEMs, including hsa-miR-4299, hsa-miR-21-5p, hsa-miR-4717, and hsa-miR-1264. The hub miRNAs have been selected for miRNA–mRNA target prediction analyses using online databases, which were based on different mathematical algorithms and scoring systems. The miRNA target prediction analysis showed SOCS7 as the target of miR-4299 [47]. Previously, the suppression of SOCS7 by other miRNAs was reported to activate STAT3 and cause its nuclear translocation [75,76]. STAT3 either promotes inflammation through the IL-6/STAT3 proinflammatory signaling axis or suppresses it through the IL-10/STAT3 axis [77]. STAT3 has been shown to promote viral replication in different viruses, including Varicella-zoster virus (VZV), hepatitis C virus (HCV), and human cytomegalovirus (HCMV) [78], and was also reported to promote viral persistence in target tissues in Zika virus [79] and Theiler’s murine encephalomyelitis virus (TMEV) [78]. CHIKV has a broad organ tropism, mainly infecting and replicating in skin and joint fibroblasts as well as in muscle fibers. The infection of these cells with CHIKV induces type I IFN responses, leading to the release of inflammatory mediators and resulting in prolonged inflammation in the infected tissue, particularly in joints. Studies have shown the involvement of the host inflammatory response, specifically the release of IL-6, as an essential mediator of the persistence of arthralgia related to CHIKV infection [50,80,81]. Therefore, it has been hypothesized that SOCS7 downregulation by hsa-miR-4299, which results in STAT3 activation and promotes inflammation through the IL-6/STAT3 axis, might be a mechanism that results in persistent viral RNA in the fibroblasts, leading to chronic CHIKV disease [47], although further experimentation is necessary to confirm this.

The PI3K-AKT-mTOR signaling pathway is involved in many cellular processes including cell differentiation, proliferation, survival, and growth [82]. AKT3, the key regulator of the PI3K-AKT-mTOR signaling pathway, has been predicted as the target of hsa-miR-4717 [47]. Activation of AKT3 is required for type I IFN production [83]. Furthermore, the inhibition of p-AKT phosphorylation has been shown to mitigate the inflammatory response in mouse synovial fibroblasts [84]. Therefore, the suppression of AKT3 by hsa-miR-4717 might be a mechanism employed by CHIKV to suppress the robust type I IFN and inflammatory responses during infection. Agrawal et al. (2020) reported the higher expression of hsa-miR-21 in response to CHIKV infection and predicted pellino E3 ubiquitin-protein ligase 1 (Peli1) as the target of this miRNA [47]. Pellino E3 ubiquitin-protein ligase 1 (Peli1) is an E3 ubiquitin ligase which suppresses the NF-kB signaling pathway by associating with the NF-κB inducing kinase (NIK) and mediating the ubiquitination and degradation of this protein [85]. The GO term and pathway analysis for Peli1 showed the correspondence of this protein to the IL-1 signaling pathway, suggesting that the downregulation of Peli1 likely suppresses the IL-1 signaling pathway and inflammation. Interestingly, the elevated expression of IL-1β in skin fibroblasts upon infection with CHIKV has been shown to activate the inflammasome via activation of caspase 1. Furthermore, the inhibition of the inflammasome by silencing caspase 1 has shown to enhance CHIKV replication in skin fibroblasts. Therefore, the activation of the inflammasome through IL-1β increases the proinflammatory response in dermal fibroblasts and contributes to the antiviral response to CHIKV [86]. Altogether, these results suggest that miR-21 targets Peli1 and suppresses IL-1 signaling, which promotes CHIKV replication during the acute phase of the disease [47].

Another study of the miRNA profile in primary human synovial cells revealed the upregulation of miR-146a and predicted TNF receptor-associated factor 6 (TRAF6), IL-1 receptor-associated kinase 1 (IRAK1), and IL-1 receptor-associated kinase 2 (IRAK2) as its targets. The knockdown and overexpression experiments confirmed the role of miR-146a as the suppressor of TRAF6, IRAK1, and IRAK2 [87,88]. TRAF6, IRAK1, and IRAK2 are signal transducers that are active in the TLR signaling pathway, and their activation leads to the activation of the NF-kB signaling pathway and inflammatory response [89]. Suppression of the inflammatory response during the infection results in elevated viral replication. Analysis of the CHIKV copy number in primary human synovial fibroblasts overexpressing miR-146a confirmed the higher replication of this virus in the presence of higher levels of miR-146a [88]. Therefore, the upregulation of miR-146a enhances CHIKV replication and survival by suppressing upstream elements of the NF-kB signaling pathway. Interestingly, CHIKV has been previously shown to impair the MDA5/RIG-I mediated activation of the NF-kB promotor [90], which leads to lower type I IFN responses and higher viral replication in the infected cells. The extensive replication of CHIKV in fibroblasts and myofibers has been reported during the acute phase of CHIKV disease. Furthermore, in vivo studies have shown that these cells survive CHIKV infection and identified them as the reservoir for persistent CHIKV RNA during the chronic phase of CHIKV disease [91]. Therefore, any alterations in the fibroblasts’ immune response to the infection that increases the CHIKV replication in these cells not only increases the viral load in the infected tissue, but also promotes the possibility of progression to chronic CHIKV disease.

The alterations in the host miRNA profile have been studied in different stages of alphavirus infection. The early events during infection determine the virus’ evasion of the cellular response and establishment of effective viral replication [43]. Computational analysis of the host miRNA profile showed significant modulations of miRNAs targeting apoptosis and JAK-STAT signaling pathways at the early stages of CHIKV infection in mouse fibroblasts [92]. miR-15 and miR-16 have been reported to promote apoptosis by negatively regulating BCL2 [93]. Furthermore, miR-23a has been reported to induce apoptosis in a caspase-dependent and -independent manner [94]. Downregulation of miR-15, miR-16, and miR-23a has been reported at the early stages of CHIKV infection in skin fibroblasts [46]. Other studies reported the elevated expression of hsa-miR-15b and hsa-miR-16 in human skin fibroblasts and miR-23a in HEK239T cells at later stages of CHIKV infection [56]. The lower expression of apoptosis-inducing miRNAs at early stages of CHIKV infection, followed by the higher expression of these miRNAs at later stages of viral infection, is in concordance with the report of delayed apoptosis during CHIKV infection [95]. Moreover, CHIKV has been reported to delay apoptosis and use apoptotic blebs to spread to the neighboring cells [95,96,97]. Therefore, the delayed apoptosis extends the replication period of the CHIKV in the infected cells and helps the virus to spread to new cells, which results in a higher viral load in the infected tissue, leading to more severe inflammation and arthritis. This trend in the modulation of apoptosis-related miRNAs might suggest a possible mechanism implicated by the virus to delay apoptosis and establish more efficient infection and spread. 

#### 2.2.2. Proviral Modulation of Viral Replication by miRNA Interaction with Viral Genome

An increasing body of evidence showed that host miRNAs can affect viral replication by directly binding to the viral RNA genome [98]. Assessing the miRNA interaction landscape on SINV, CHIKV, EEEV, WEEV, and VEEV RNA genomes illustrated the sequestration of a broad range of host miRNAs by the viral genome, suggesting these miRNAs play a role in viral replication [99]. Notably, the interaction of the hematopoietic-specific miRNA, miR-142-3p, with the 3′ non-translated region (NTR) of the EEEV genome has been shown to prevent the translation of the viral genome, leading to restricted EEEV replication in hematopoietic cells and a suppressed innate immune response to the infection. The myeloid lineage tropism caused by the EEEV genome interaction with miR-142-3p allows the virus to avoid the type I IFN response and enter the central nervous system (CNS) with minimal interruption from the mammalian host immune system [100]. The further study of the 3′ NTR sequence in the EEEV genome identified four conserved binding sites for miR-142-3p, which were also found in the 3′ NTR of the WEEV genome. The interruption of miR-142-3p binding sites on the genome of EEEV and WEEV was associated with higher viral replication, confirming the role of this miRNA in the suppression of these viruses’ replication [101]. Moreover, the 3′ NTR binding site of miR142-3p has been recognized as an essential sequence for efficient EEEV infection in mosquito cells, suggesting this interaction as a mechanism for the positive selection of the miR142-3p binding site during vector-to-vertebrate host transmission [100]. Therefore, the interaction of miR-142-3p with the EEEV genome is essential for immune evasion and the successful establishment of viral infection.

Although miRNA-mediated regulation regularly causes gene suppression, examples of elevated viral replication induced by miRNA binding to the viral genome exist [102]. A particular miRNA interaction with the 5′ UTR of the viral genome, which results in Argaunat2 (Ago2)-mediated viral translation initiation, has been indicated previously [103]. The study of Ago2 inhibition/deletion using the Ago2 inhibitor, acriflavine (ACF), or Ago2^−/−^ cells showed decreased viral replication, structural protein expression, and viral titer in VEEV, EEEV, and WEEV, indicating the positive effect of Ago2 and RNAi machinery in New-World alphavirus replication [104]. The positive regulatory effect of miRNA on alphavirus replication was particularly studied in SINV, where sixteen miRNAs were identified with positive regulatory effects on SINV replication. Notably, the neuron-specific miRNA, miR124-3p, by binding to its binding sequence in the E1 coding region of the SINV genome, increased capsid protein expression and viral titer in human differentiated neuronal cells. The positive regulatory effect of miR-124-3p on alphavirus structural protein expression and viral titer has been further confirmed in CHIKV-infected human cells [105].

### 2.3. Mosquito miRNAs in Response to Alphavirus Infection

The transmission of the alphaviruses to their vertebrate hosts requires the establishment of an efficient viral infection in the mosquito vector. Female mosquitos acquire alphavirus infection by feeding on the blood of a viremic vertebrate host. The virus then replicates in the mosquito’s midgut and disseminates to the salivary glands, leading to transmission to another vertebrate host during the next bloodmeal [106]. Similar to the viral infection in vertebrates, the mosquito cells use cellular pathways to control viral infection, and the virus alters the transcriptional profile of the vector cells. In mosquito cells, RNAi pathways are the major parts of the antiviral immune response to viral infection [87]. The alterations of RNAi pathways in response to infection affect the viral replication by regulating the host factors or direct miRNA–viral RNA (vRNA) interaction. In this section, we review the role of arthropod miRNAs in response to alphavirus infections.

Mayaro virus (MAYV) is a part of the Semliki complex within the Alphavirus genus, which causes human diseases characterized by fever, skin rash, and arthritis [107]. The study of the transcriptomic and small RNA response to MAYV in Anopheles stephensi identified significant modulations of a set of eight miRNAs, from which three were novel and five were known. In response to MAYV infection, the five known miRNAs, aga-miR-286b, aga-miR-2944a, aga-miR-2944b, aga-miR-307, and aga-miR-309, were modulated as a group, as they were suppressed between 2–7 d.p.i. and enriched between 7–14 d.p.i. [108]. Modulations of miRNAs as a group may mean that they are co-regulating a set of genes in response to the infection. Gene ontology (GO) analysis has shown modulations of proteases between 2–7 days p.i. that suggest the activation of the Toll pathway. In addition, the pathways related to autophagy and apoptosis were suppressed at 14 d.p.i. [108]. These results highlight the mosquito antiviral response at both early and late time points of infection. 

The differential expression of a set of eight miRNAs in response to CHIKV infection has been reported in Aedes albopictus, from which four miRNAs, including miR-100, miR-305-3p, miR-283, and miR-927-5p, were upregulated. Pathway prediction analysis indicated immune-related pathways, metabolic pathways, and pathways related to viral entry as targets of upregulated miRNAs. miRNA–mRNA target analysis identified the protein tyrosine phosphatase SHP2, ERK1/2, and ubiquitin fusion degradation protein as targets of upregulated miRNA [109]. The importance of ERK signaling in the antiviral response in insects during arbovirus infection has been highlighted previously. Reports demonstrated that the suppression of ERK is associated with elevated arbovirus infection [110,111]. Therefore, the upregulated miRNAs possibly increase viral infection by suppressing ERK. Further pathway analysis using the target genes of downregulated miRNAs (miR-1000, miR-2b, miR-2c, and miR-190-5p) post-CHIKV infection identified the ribosome pathway as the target of these miRNAs [109]. A study of the miRNA profile in CHIKV-infected Aedes aegypti mosquitos and the Ae. aegypti-derived cell line (Aag2 cells) confirmed the modulation of miR-2b and miR-100 in response to infection. miR-2b interacts with the 3′UTR of the ubiquitin-related modifier (URM) transcript and decreases its expression [112]. URM suppression inhibits thiolation of tRNAs [113], and subsequently, may decrease CHIKV replication. Therefore, the downregulation of miR-2b in CHIKV infection leads to increased CHIKV replication. miR-100 inhibited the production of human cytomegalovirus infectious progeny by silencing components of the mTOR pathway [114]. While the antiviral role of miR-100 has been reported in HCMV, inhibiting miR-100 did not modulate the CHIKV viral copy number in Ae. aegypti [112]. It is noteworthy that the genome copy number does not provide information about the number of infectious virions. Thus, further studies are required to illustrate the role of miR-100 in CHIKV infection. Another consideration is the opposite patterns of miR-2b and miR-100 expression in two CHIKV-infected Aedes vectors. While CHIKV infection increased miR-100 and decreased miR-2b expression in Ae. albopictus cells, the expression of miR-100 decreased, and miR-2b increased in Ae. aegypti. This differential modulation of the same miRNAs in response to the same viral infection might represent the species-specific antiviral response in Aedes mosquitos. 

The study of the small RNA profile during the persistent CHIKV infection (14 d.p.i.) in ovaries and carcasses of Ae. albopictus mosquitoes showed the differential expression of 14 miRNA, from which 13 had homologs in Ae. Aegypti [115]. Among the differentially expressed miRNAs, eight, including aal-miR-210, aal-miR-124, aal-miR-1000, aal-miR-219, aal-miR-932, aal-miR-981b, aal-miR-193, and aal-miR-285, were upregulated, whereas six, including aal-miR-2941, aal-miR-7, aal-miR-316, aal-miR-9b, aal-miR-new6, and aal-miR-1891, were downregulated. Interestingly, except for miR-1000, the rest of the modulated miRNAs had not been previously reported in CHIKV-infected Ae. Albopictus [115]. Furthermore, the suppression of miR-1000 was already reported in CHIKV-infected Ae. Albopictus at 24 h.p.i. [109], whereas it was upregulated in the ovaries of CHIKV-infected Ae. albopictus at 14 d.p.i. [115]. The KOBAS pathway analysis predicted 12 pathways, including Toll and IMD pathways, as the targets of upregulated miRNAs in the ovaries of CHIKV-infected Ae. albopictus at 14 d.p.i., suggesting the suppression of these antiviral pathways during persistent infection with CHIKV [115]. Moreover, the vertical transmission of CHIKV has been reported in Ae. albopictus previously. This phenomenon is related to the transfer of CHIKV from the infected parent mosquito to its offspring in the ovary during the oviposition [116]. Therefore, these results suggest that a specific miRNA profile during persistent CHIKV infection may suppress the immune response to the infection and help establish viral persistence in the ovaries, leading to the vertical transmission of CHIKV.

Studies of the miRNA repertoire post-Arbovirus infection in Aedes mosquitos have reported the alteration of miR-2944b in response to CHIKV and Zika virus infections [109,117]. The modulations of miR-2944b were also reported in response to MAYV in An. stephensi [108]. Later studies have shown that miR-2944b-5p binds to the 3′ UTR of the CHIKV genome, leading to decreased viral replication in Aag2 cells and Ae. aegypti mosquitos. Although decreased viral replication by miR-2944b-5p is an antiviral effect, it might result in a suppressed type I IFN response, leading to virus evasion from the cellular response [101]. miRNA–mRNA target analysis predicted vacuolar protein sorting-associated protein 13 (VPS-13) as the host target of miR-2944b-5p [118]. VPS-13 regulates membrane morphogenesis and maintains the mitochondria function in other systems [119,120]. Many viral infections impair mitochondria, leading to changes in the mitochondria membrane potential (MtMP). The measurement of MtMP effectively identifies damaged mitochondria [121]. MtMP measurement in CHIKV-infected Aag2 cells demonstrated the maintenance of mitochondrial integrity, suggesting the role of miR-2944b-5p and its target gene VPS-13 in regulating the mitochondrial integrity. These results suggest that miR-2944b-5p exerts its proviral effect in mosquito cells by restricting the viral replication and maintaining mitochondrial integrity through its target gene VPS-13, which results in persistent infection [118].

Ross River virus (RRV) is another member of the Old-World alphaviruses. The study of the small RNA profile in the midgut and fat body of the Ae. aegypti infected with RRV identified 14 differentially expressed miRNAs post-infection, among which miR-9b-5p was upregulated in both the midgut and fat body at 2 d.p.i. [122]. In human cells, miR-9b has been reported to promote viral replication through the suppression of MCPIP1, which is involved in innate host defense against influenza A virus infection [123]. In mosquitoes, the epithelial cells of the midgut act as the first line of defense against viral acquisition through a bloodmeal by synthesizing antimicrobial peptides (AMPs) and reactive oxygen species (ROS). In addition, the fat body is the primary site of humoral response in mosquitos, which affects the viral infection by producing and releasing AMPs [124]. Although the role of miR-9b in alphavirus infection has not been investigated, the increased expression of this miRNA in the midgut and fat body of Ae. aegypti at early time points post-RRV infection may suggest the role of miR-9b in alphavirus replication and suppression of the mosquito immune response to the infection. Among miRNAs modulated in response to RRV infection, miR-275-5p showed the most suppression and miR-989-3p was the most upregulated at 2 d.p.i. in the Ae. aegypti fat body [122]. Previously, the upregulation of miR-989 in the midgut of An. gambiae infected with Plasmodium bergei was reported [125]. Other studies have reported the modulations of miR-989-3p in response to West Nile virus (WNV) infection in Ae. albopictus and Culex quinquefasciatus, [126] and Ae. aegypti infected with DENV [127] or ZIKV [117]. miRNA–mRNA target and pathway analysis showed genes involved in zinc finger and MAPK signaling pathways as the target of miR-9b-5p and miR-989-3p, suggesting the role of these miRNAs in the modulation of the mosquito immune response to alphavirus infection. 

Although miRNAs are generally considered to be intracellular, their presence in biofluids has been reported [128,129]. The mosquito saliva contains a complex repertoire of secretory proteins and factors that affect pathogen transmission and disease establishment through the modulation of the immune response at the site of infection [130,131,132]. The study of the miRNA composition in the saliva of Ae. aegypti and Ae. albopictus identified a total of 103 miRNAs in the saliva of these mosquitos [133]. Fiorillo et al. (2022) reported the expression of 208 miRNAs in saliva and salivary glands of uninfected and CHIKV-infected Ae. aegypti and confirmed a 92–94% similarity between the top 50 saliva miRNAs with the other report, suggesting a conserved miRNA expression pattern in the saliva of Ae. aegypti [134]. The investigation of the role of the modulated miRNAs in CHIKV infection using miRNA inhibitors showed lower CHIKV titer in aae-mir-184-, aae-mir-375-, and aae-mir-2490-inhibited Aag-2 and BHK-22 cells, suggesting the role of these miRNAs in CHIKV replication in both mosquito and mammalian cells [133]. Interestingly, the higher expression of miR-184 in response to proinflammatory cytokine IL-22 has been reported in human keratinocytes, corresponding with the downregulation of Ago2 expression [135]. The Ago2 protein is a component of the RISC complex, which is involved in the RNAi defense against viral infection in mosquitos [87]. Therefore, the modulation of miR-184 in the saliva of the CHIKV-infected mosquitos may alter the host immune response by modulating Ago2 at the site of infection. miR-375 has been predicted to target the regulators of the Toll immune pathway, RNA-editing ligase 1 (REL1), and cactus. The upregulation of cactus has been reported in response to the miR-375 mimic injection to Ae. aegypti mosquitos [136]. As cactus is the inhibitor of the NF-kB transcription factor activation, the decreased expression of aae-mir-375 in the saliva of CHIKV-infected Ae. aegypti and Ae. albopictus increases the activation of the NF-kB transcription factor by reducing the expression of cactus, which controls viral replication at the infection site. The inhibition of aae-mir-2490 also decreased the CHIKV titer in Ae. aegypti and mammalian cells [133]. Aae-mir-2490 targets metalloprotease m41 ftsh and increases its expression in Ae. aegypti cells and mosquitoes after Wolbachia infection (a bacteria that commonly infects insects and can modulate viral replication), which results in higher Wolbachia replication [137]. Furthermore, the inhibition of aae-mir-2490 in Aag-2 cells decreased Wolbachia replication [137]. Altogether, these results suggest that the miRNA profile of the mosquito saliva may affect arbovirus replication and infection by affecting the host gene expression at the site of infection. 

## 3. Long Non-Coding RNAs (lncRNA) in Alphavirus Infection

Alphaviral infection leads to the alteration of the host transcriptome and activation of the host antiviral response, and involves many signaling pathways. Recent studies have shown the alteration of lncRNA expression in response to viral infection and confirmed the critical regulatory role of these RNAs at the host–pathogen interface. lncRNAs modulate viral infection at different levels. The antiviral lncRNAs exert their effect by directly inhibiting the viral infection or stimulating the antiviral immune response, whereas the proviral lncRNAs improve viral replication or suppress the antiviral response [138,139,140,141]. Besides host lncRNAs, viral lncRNAs can also control the viral life cycle and infection by regulating the transcription or translation of the host or viral genes [142]. In this section, we review the role of lncRNAs in alphavirus infection. 

Although CHIKV is known to cause joint and muscle pain, rare cases of encephalitis have been reported in CHIKV-infected patients [143]. Brain microvascular endothelial cells are the main components of the blood–brain barrier (BBB), representing a barrier to virus dissemination into the brain through the blood route [144]. Therefore, studying the cellular response to CHIKV infection in cells involved in the BBB may elucidate the mechanism by which CHIKV infects the CNS. One study has identified 9 potentially antiviral and 21 putative proviral lncRNAs in CHIKV-infected human brain microvascular endothelial cells (HBMECs). Further experiments focused on an unknown cytoplasmic lncRNA, later named antiviral lncRNA prohibiting human alphaviruses (ALPHA), confirming the antiviral role of this alphavirus-specific lncRNA and validating that the antiviral function of ALPHA is independent of type I IFN signaling. The further assessment of the ALPHA mechanism of action has shown the direct interaction of ALPHA exon 1 with nsP1 within the CHIKV genome, suggesting that ALPHA inhibits CHIKV replication through direct interaction with the viral genome. Although the induction of ALPHA was identified in several alphaviruses, including CHIKV, ONNV, SINV, and MAYV, the depletion of ALPHA only increased viral yields in CHIKV and ONNV-infected cells, showing the virus-specific function of this lncRNA. Although this report illustrated the inhibitory effect of the lncRNA ALPHA on CHIKV and ONNV replication, the cell type-specific expression of the lncRNAs may limit this effect to HBMECs. Therefore, further experiments are needed to investigate the expression level and effect of ALPHA on CHIKV replication and virion formation in other CHIKV target cells, particularly fibroblasts. Additionally, the role of other potential anti- and proviral lncRNAs in CHIKV infection in HBMECs remains to be investigated. 

## 4. Small Nucleolar RNA (snoRNA) in Alphavirus Infection

Small nucleolar RNAs (snoRNAs) are a class of non-coding RNAs that are 60–300 nt long [143]. The snoRNAs are classified as either C/D box or H/ACA box snoRNAs, SNORD and SNORA, respectively. These highly expressed non-coding RNAs are localized in the nucleus and act as the scaffold to form a small nucleolar ribonucleoprotein (snRNP) complex. snoRNAs guide the snRNP complexes to target RNAs, particularly rRNAs and small nuclear RNAs (snRNAs), which results in nucleotide modifications of cellular RNAs [145,146,147,148,149,150,151]. Although mostly found in the nucleus, studies have shown the presence of SNORDs in the cytoplasm in response to cellular stimuli [152,153]. Other studies revealed the interactions between snoRNAs and pre-mRNAs and demonstrated the further processing of snoRNAs into miRNAs and other shorter types of RNA, suggesting the role of snoRNAs in alternative pre-mRNA splicing and the modulation of gene expression [148,154,155,156]. 

A growing body of evidence showed the alteration of host snoRNAs in response to infections with RNA and DNA viruses and illustrated the role of these host factors in viral replication and infection [157,158,159,160,161,162]. For example, a gene-trap study using twelve viruses and various cell lines identified the genes whose disruption led to resistance to lytic infections. Interestingly, 83 SNORDs/SNORAs were identified as the transcripts of the disrupted genes. Knockdown studies using siRNAs targeting candidate SNORDs/SNORAs demonstrated the role of these snoRNAs in viral infection, including cowpox virus (CPV), dengue virus (DENV), influenza A virus (FLU), human rhinovirus 16 (HRV16), herpes simplex virus 2 (HSV2), and respiratory syncytial virus (RSV). Notably, the suppression of eight snoRNAs inhibited the replication of three or more viruses in the treated cell lines [157]. Another study showed that SNORD126 promotes HCV infection by increasing phosphorylated AKT levels, which results in the enhanced expression of the HCV key entry factor, claudin-1 (CLDN-1) [159]. Altogether, these studies suggest that C/D box-type snoRNAs are supporting viral replication and entry. 

The study of the snoRNA expression signature in CHIKV-infected HEK293T cells showed upregulation of a total of 48 snoRNAs at 12 and 24 h post-infection, from which 16 were C/D box type, including SNORD29, SNORD38, SNORD44, SNORD76, SNORD78, SNORD83A, and SNORD83B, which were previously called U29, U38, U44, U76, U78, U83A, and U83B [56]. Interestingly, the disruption of the genes encoding these snoRNAs has been reported in cells that survived viral lytic infections. For example, the disruption of the genes encoding SNORD44, SNORD76, and SNORD78 has been observed in cell lines that survived the lytic infections with CPV, human rhinovirus 2 (HRV2), HRV16, HSV2, reovirus (REO), and RSV. Similarly, the disruption of SNORD83A- and SNORD83B-encoding genes has been observed in the cell lines that survived lytic infection with CPV and HRV6 [157]. In addition, Murray et al. (2014) reported that the disruption of the gene encoding SNORD29 in the cell lines survived lytic infection with HRV16 and HSV2. The knockdown (KD) experiments using siRNAs targeting nine SNORDs showed the inhibition of viral replication in the treated cells, suggesting the role of C/D box snoRNAs in viral replication. Particularly, the replication of CPV, FLU, HRV16, and HSV2 was inhibited in SNORD29 KD cells [157]. These results suggest that upregulated C/D box snoRNAs after CHIKV infection may support CHIKV replication. However, the role of these snoRNAs in CHIKV replication and infection remains to be investigated. 

## 5. Discussion

Most studies of ncRNAs in alphavirus infection have focused on miRNAs in CHIKV infection, and have identified a large number of these RNAs that have roles in pro- and antiviral responses. Infection with CHIKV causes acute and chronic polyarthralgia and/or polyarthritis. Upon infection, CHIKV replicates in dermal fibroblasts, which is followed by the dissemination of the virus into different organs, specifically joints, and muscles. CHIKV replication in fibroblasts and endothelial cells elicits the type I IFN response to the infection. Activation of the IFN response and its downstream signaling pathways leads to induction and secretion of the proinflammatory mediators, which is followed by infiltration of immune cells into the infected organs, especially joints, and inflammation leading to arthritis [49,163,164]. The chronic stage of CHIKV disease is characterized by musculoskeletal pain lasting for months or years. The chronic phase of the disease is not the result of chronic infection, as the infectious virus is not isolated from the patients. However, the persistent viral RNA was detected in joint tissues from the patients [49]. Studies identified the myofibers and dermal and muscle fibroblasts as the reservoir for persistent viral RNA during the chronic stage of CHIKV disease [91]. Moreover, studies have shown the involvement of the host inflammatory response, specifically the release of IL-6 and IL-1β, as important mediators of chronic CHIKV disease [81]. Thus, the modulation of cellular immune responses which decrease the viral RNA load in CHIKV target cells, particularly fibroblasts, and lower the inflammatory response to the infection may alleviate the joint pain in the acute phase and lower the progression to the chronic phase of CHIKV disease. As non-coding RNAs are regulators of mRNAs and pathways, the study of their function in response to infection can provide beneficial information about the host–virus interaction. Although the modulation of miRNAs has been reported in response to the CHIKV infection in fibroblasts, and the target pathways have been predicted for the miRNAs, in most cases, the mechanism by which these miRNAs control or support the viral infection has not been fully understood. Therefore, further investigations are needed to clarify the role of these miRNAs in CHIKV infection. The miRNA/lncRNA response of mammalian cells to CHIKV infection is summarized in Figure 1. A limited number of studies have analyzed other types of ncRNAs, including snoRNA and lncRNA, responses in CHIKV infection. These studies have highlighted the importance of non-coding RNA in CHIKV infection in both human and insect vector cells. Comparatively, very little is known about ncRNA responses to other alphaviruses, including non-CHIKV Old-World alphaviruses and the encephalitic New-World alphaviruses such as VEEV, EEEV, and WEEV. In addition, ncRNA types other than miRNA are less well characterized in alphavirus infection. Therefore, future research should include studies on understudied alphaviruses and ncRNA types. Table 1 summarizes the types of ncRNAs that have been characterized in different alphaviruses.

Understanding the role of non-coding RNAs in viral infection is of increasing interest as it may result in the identification of novel biomarkers and/or non-coding RNA-based therapeutic targets against viral infections. An ideal biomarker should meet criteria such as accessibility, high specificity, and sensitivity. The specificity for the disease, presence and stability in body fluids, and the existence of established tools for measurement make non-coding RNAs, such as miRNAs, lncRNAs, and snoRNAs, good candidates for biomarker research. The application of non-coding RNAs, e.g., miRNAs, lncRNAs, and snoRNAs, as disease diagnostic/prognostic biomarkers, has been explored in numerous non-infectious diseases, e.g., various cancers [165,166,167,168,169,170,171]. In viral infections, the study of miRNAs in body fluids identified potential biomarkers for Ebola virus [172], HIV-1 [173], influenza A virus (IAV) [174], and rhinovirus [175] infections, which was further reviewed by [176]. Furthermore, the lncRNAs in biofluids have been reported as diagnostic markers for chronic hepatitis B virus (HBV) [177] and HBV-related cirrhosis [178], and as a prognostic marker for visual acuity in patients with acute retinal necrosis caused by herpes simplex virus type-1 (HSV-1) [179]. In addition, the circulating snoRNAs have been reported as potential prognostic markers for COVID-19 severity [180]. These studies prove the likelihood of using biofluids’ non-coding RNAs in response to viral infections for the development of diagnostic/prognostic tools against viral infections. This is particularly important for the correct diagnosis of viruses that cause non-specific symptoms at the early stages of the disease, which may later progress into severe illnesses. Although the expression signature of miRNAs has been studied in response to multiple alphaviruses, to the best of our knowledge, there is no report of non-coding RNAs in biofluids from patients. The alphaviruses cause flu-like symptoms at the early stages of the disease, which may progress into CNS infection and encephalitis in encephalitic alphaviruses or arthritis in atherogenic alphaviruses. Therefore, the study of the non-coding RNA signature in patients’ biofluids may help with recognition of biomarkers for early alphavirus diagnosis or identification of prognostic biomarkers which may further change the outcome of the disease. 

Another advantage of identifying non-coding RNAs and their role in viral infection is the potential development of non-coding RNA-based drugs. The differential expression of miRNAs and lncRNAs in the disease condition and their ability to regulate gene expression and pathways raise the potential for miRNA- and lncRNA-based therapeutic development [181,182,183]. The genome of viruses can be bound by miRNAs and lncRNAs, which results in negative or positive regulation of the viral replication [98,184]. In addition, the modulation of the viral infection of miRNA and lncRNA expression alters the host mRNA expression, causing pro- or antiviral effects [184,185,186,187]. Therefore, the replacement or inhibition of miRNAs and lncRNAs could be effective against viral infections. Among the multiple miRNA-based drugs in clinical trials, two are against the hepatitis C virus (HCV) [182]. The interaction of liver-specific microRNA, miR-122, with the 5′ UTR of the HCV genome is essential for HCV replication in hepatocytes [188]. Miravirsen, a locked nucleic acid (LNA), suppresses HCV replication by inhibiting miR-122 and has successfully lowered HCV RNA under the detectable level and cleared phase II clinical trials [189,190]. The serial passage in the presence of Miravirsen did not cause mutations in the miR-122 binding site. However, the recurring mutation in position 4 of the HCV 5′ UTR resulted in lower drug effectiveness, possibly by stabilizing the RNA. In the same study, a C3U mutation was reported in the HCV 5′ UTR from patients with a viral rebound after therapy when Miravirsen was completed [191]. Rg-101 also decreased HCV RNA to an undetectable level in patients by antagonizing miR-122 in hepatocytes. However, the application of Rg-101 has been suspended in the phase 2 clinical trial due to adverse reactions in patients [192,193]. Despite the rapid progress in the non-coding RNA therapeutic field, no lncRNA-based drug has yet entered into clinical trial. However, a few preclinical studies have shown the effectiveness of lncRNA-based drugs in mouse models [183]. Preclinical studies demonstrated that the successful suppression of MALAT1 and SAMMSON by antisense oligonucleotides (ASO) triggered cancer cell death in xenograft mouse models of multiple myeloma and melanoma, respectively [194,195], suggesting the potential use of ASOs for targeting lncRNAs as a therapeutic. Because of the ability to regulate cellular pathways, lncRNAs have the potential to be used as effective drugs. However, the large size of lncRNAs makes the delivery challenging and elicits an immune response to therapy. For example, the pharmacological delivery of the full-length NRON successfully inhibited bone resorption in the mouse model of osteoporosis, but caused side effects in mice due to the activation of the immune response [196]. The delivery of the lncRNA’s functional motif may resolve the problems of a challenging delivery and activation of the immune response. For example, the delivery of the NRON functional motif efficiently inhibited bone resorption in the osteoporotic mouse model, whereas it did not activate the immune response [196]. 

Could non-coding RNA-based drugs efficiently control alphavirus infection? The study of the miRNA interaction landscape on the genome of multiple alphaviruses has revealed the sequestration of a broad range of miRNAs by alphaviruses’ genomes. However, the role of these miRNAs in viral replication still needs to be investigated [99]. Another study revealed the positive role of Ago2 in encephalitic alphavirus replication and infection, suggesting the role of RNAi machinery in alphavirus replication [104]. Furthermore, multiple studies have shown the direct interaction of miRNAs or a lncRNA with alphaviruses’ genomes and identified the alteration of viral replication and titer due to the interaction with these non-coding RNAs [100,101,105,184]. On the other hand, the modulation of certain host miRNAs and lncRNAs by viral infection can cause pro- or antiviral effects through the alteration of host immune pathways. Many studies have shown modulations of host miRNAs in response to alphavirus infections and predicted host immune pathways as the targets of these miRNAs [46,47,56,70,92]. These results show the possible regulation of alphavirus infection by miRNAs/lncRNAs and indicate the likelihood of controlling alphavirus infection by miRNA-/lncRNA-based drugs. However, there are not a lot of data available about the dysregulation and role of non-coding RNAs in alphavirus infection. For example, only one report exists on the modulation of lncRNAs in response to alphavirus infection [184]. Similarly, there is only one record of snoRNA alteration in alphavirus infection [56]. In addition, to the best of our knowledge, the alterations and roles of circular RNA (CircRNA), another group of regulatory RNAs with potential to be used as therapeutics [197], in alphavirus infection has not been studied. Altogether, even though the current knowledge shows that non-coding RNAs play important roles in alphavirus infection, further research is needed to identify the non-coding RNA candidates and determine the mechanism of their roles in alphavirus disease. Furthermore, the transition from laboratory findings to clinical applications is challenging and requires extensive preclinical and clinical studies.

## Figures and Tables

**Figure 1 viruses-15-00562-f001:**
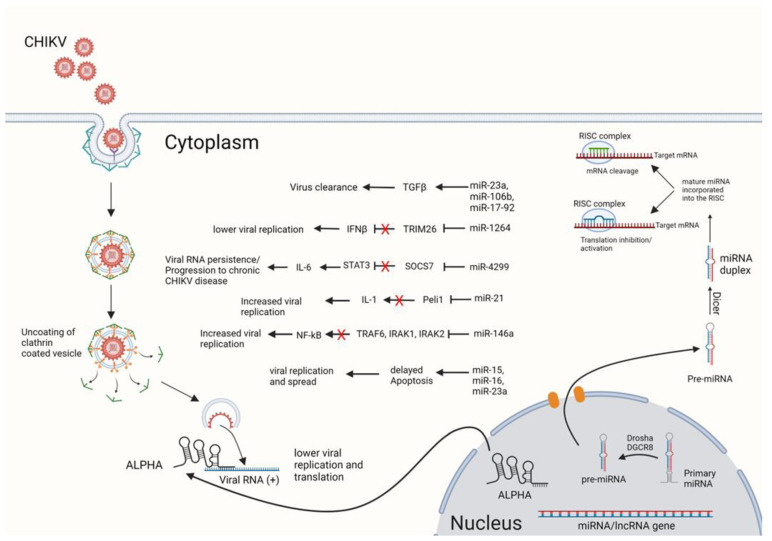
Summary of miRNA/lncRNA response in CHIKV-infected mammalian cells. CHIKV infection modulates miRNA/lncRNA expression in mammalian cells, such as fibroblasts. Alteration of immune-related genes’ transcript levels or translation of their mRNAs results in modulation of the immune response to CHIKV infection, leading to pro- or antiviral effects. The lncRNA ALPHA decreases CHIKV replication and protein translation by directly binding to the viral genome. Image created using BioRender.

**Table 1 viruses-15-00562-t001:** ncRNAs in alphavirus infection.

ncRNA Type	Target	Alphaviruses Analyzed
lncRNA	Virus	CHIKV, ONNV
miRNA	Host, Virus	CHIKV, VEEV, EEEV, WEEV, SINV, MAYV, RRV
snoRNA	Host	CHIKV
circRNA	NR	NR
piRNA	NR	NR

NR, not reported.

## Data Availability

Not applicable.

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
