# Peer review of "The Host Non-Coding RNA Response to Alphavirus Infection"

_viruses, 2023, doi:10.3390/v15020562_

Round 1

Reviewer 1 Report

This manuscript on host non-coding RNA response to alphavirus infection reviews the current literature on the topic. This is an interesting topic and compilation of the literature is useful, a thoughtful review would be more welcome. The findings are presented as reportage of the literature with little synthesis of the information or interpretation of the importance of this aspect of host cell biology. Specific problems include:

1.     Problems related to carelessness: Misspellings, un-generated citations, unexplained insertions of what look like page numbers, etc.

2.     Two sections labeled “Discussion”

3.     Lack of summary figures or graphs e.g. maybe synthesize the data on CHIKV infection.

4.     Cell type-dependence of these responses and importance are unclear. Role in RNA persistence?

Reviewer 2 Report

Manuscript viruses-2184737 by Bradfute et al. describes the roles of ncRNA in host cells and vector cells in response to Alphavirus infection, especially miRNA. The manuscript describes that upon viral infection, how host cells interfere with viral replication by differentiating miRNA profile, either up-regulation or down-regulation levels of miRNA to act on relevant pathways including immune, apoptosis, inflammation and that how viruses take advantage of host or viral miRNA profile as a method to benefit viral persistence, immune evasion and infection by acting on the relevant pathways. The manuscript also describes some roles of lncRNA and snoRNA in viral infection, however further studies were needed to discover more of such RNAs and their roles.

The manuscript is a good review and contains highly valuable information for the field, to include contents describing ncRNA to regulate multiple stages of viral proliferation - virus transmission, persistence, genome replication, and release, and summarizing ncRNA to combat viral infection. Studies of ncRNA could be critical to develop new antiviral strategies by targeting specific ncRNA as diagnostic biomarker or as substrate for antiviral discovery.

I only have minor comments:

In line 208, has-miR-1264 appeared twice, should it be miR-146a.

The title of section 4 needs to be corrected as this is not Discussion

Round 2

Reviewer 1 Report

Grammar is awkward in several places, but the authors have adequately addressed the scientific issues.  Figure is very helpful. This will be a useful review.